# Robotic Medtronic Hugo™ RAS System Is Now Reality: Introduction to a New Simulation Platform for Training Residents

**DOI:** 10.3390/s23177348

**Published:** 2023-08-23

**Authors:** Loris Cacciatore, Manuela Costantini, Francesco Tedesco, Francesco Prata, Fabio Machiella, Andrea Iannuzzi, Alberto Ragusa, Noemi Deanesi, Yussef Rashed Qaddourah, Aldo Brassetti, Umberto Anceschi, Alfredo M. Bove, Antonio Testa, Giuseppe Simone, Roberto Mario Scarpa, Francesco Esperto, Rocco Papalia

**Affiliations:** 1Department of Urology, Fondazione Policlinico Universitario Campus Bio-Medico, 00128 Rome, Italy; loris.cacciatore@unicampus.it (L.C.); francesco.tedesco@unicampus.it (F.T.);; 2Department of Urology, IRCCS Regina Elena National Cancer Institute, 00144 Rome, Italy

**Keywords:** Hugo RAS, residents, robotic platform, robotics training, urology, virtual simulator

## Abstract

The use of robotic surgery (RS) in urology has grown exponentially in the last decade, but RS training has lagged behind. The launch of new robotic platforms has paved the way for the creation of innovative robotics training systems. The aim of our study is to test the new training system from Hugo™ RAS System–Medtronic. Between July 2020 and September 2022, a total of 44 residents from urology, gynaecology and general surgery at our institution participated in advanced robotic simulation training using the Hugo™ RAS simulator. Information about sex, age, year of residency, hours spent playing video games, laparoscopic or robotic exposure and interest in robotics (90.9% declared an interest in robotics) was collected. The training program involved three robotic exercises, and the residents performed these exercises under the guidance of a robotics tutor. The residents’ performance was assessed based on five parameters: timing, range of motion, panoramic view, conflict of instruments and exercise completion. Their performance was evaluated according to an objective Hugo system form and a subjective assessment by the tutor. After completing the training, the residents completed a Likert scale questionnaire to gauge their overall satisfaction. The rate of the residents’ improvement in almost all parameters of the three exercises between the first and the last attempts was statistically significant (*p* < 0.02), indicating significant progress in the residents’ robotic surgical skills during the training. The mean overall satisfaction score ± standard deviation (SD) was 9.4 ± 1.2, signifying a high level of satisfaction among the residents with the training program. In conclusion, these findings suggest that the training program utilizing the Hugo™ RAS System is effective in enhancing robotic surgical skills among residents and holds promise for the development of standardized robotics training programs in various surgical specialties.

## 1. Introduction

In the last decades, robot-assisted surgery (RAS) has gained popularity in several surgical fields, such as urology [1], gynaecology and general surgery, due to its faster learning curve (LC) than laparoscopy [2] and the availability of ancillary technologies that may improve perioperative outcomes, despite the higher costs per procedure when compared to other approaches [3]. Furthermore, given the wide diffusion of robotic platforms, the interest in standardized simulation-based training programs has grown over time to ensure surgeons’ dexterity with robotic skills and patients’ safety during surgery. Specifically, laparoscopic/robotic surgery is a current reality in which simulation programs should play a key role in preparing the surgeon for challenging situations, minimizing the risks as much as possible. Actually, there are several validated laparoscopic training programs, such as the Fundamental Laparoscopic Skills (FLS) course in the USA [4]. However, the lack of structured and standardized robotics training courses represents an unsolved issue in the field of RAS [5,6,7,8].

To date, several modalities of RAS curricula have been proposed, such as the daVinci Technology Training Pathway and Fundamentals of Robotic Surgery (FRS), the Fundamental Skills of Robot-Assisted Surgery (FSRS) training program and the Robotics Training Network curriculum [9], although the availability of hands-on training provided by these programs is not congruous. Furthermore, the introduction of virtual reality (VR) simulators and artificial intelligence (AI) has provided new opportunities for surgical trainees. Current simulators allow trainees to both familiarize themselves with the console settings and to improve their psychomotor and basic procedural skills [10]. Indeed, a recent study compared the da Vinci Trainer™ (dVT) and “daVinci Skills Simulator” (dVSS), in trainees with varying robotics experience [11]. Similarly, another study reported improvement using the Versius Surgical simulator, describing increased skills in all surgical specialties in trainees with previous robotics experience [12].

In this context, the introduction and rapid spread of the new Hugo RAS™ system (Medtronic, Minneapolis, MN, USA), since the achievement of the CE (Conformité Européenne) mark of approval for gynaecological and urological procedures in adult patients at the beginning of 2022, has rekindled the interest in providing a new robotic simulation platform. The aim of this study was to test and validate a new training system using a novel promising robotic platform.

## 2. Materials and Methods

### 2.1. Participants

From July 2022 to September 2022, 53 residents of different specialties from our institution (urology, general surgery and gynaecology) were enrolled in an advanced integrated robotic simulation training course using the Hugo™ RAS System. Residents with previous Hugo™ RAS System simulator experience (n = 9) from other institutions were excluded from our training. During the enrolment, the following information was collected for each resident:-Sex;-Age;-Year of residency (ranging from 1 to 5);-Hours spent playing video games per day (categorized as 0 h/day, <3 h/day or >3 h/day);-Previous laparoscopic exposure (categorized based on roles: observer (>50 procedures), assistant or first surgeon);-Robotic exposure (categorized based on roles: observer (>50 procedures), bed-assistant or console surgeon);-Interest in robotic surgery (measured on a Likert scale from 0 to 4 points).

The training course was composed of two main sections: an introductory section, in which the specialized “robotics tutor” provided an overview of the console setting, the robotic arm components and the docking and undocking functions of the Hugo™ RAS System. Afterwards, the main step of the training was based on intraoperative tasks. Following a demonstration and description of the exercises by the mentor, each resident performed all of the exercises three times. The goal was to assess the individual learning curve (LC) for each resident.

### 2.2. Training Session

The training course focused on three specific tasks using the Hugo™ RAS System: “endoscope targeting” (Figure 1), “cut and coagulation” (Figure 2) and “suturing skills” (Figure 3), with the purpose of analyzing the trainees’ visual, accuracy and dexterity capacities, respectively. In more detail, during each attempt, we focused our attention on five standardized parameters:-Timing (parameter A): The time taken by the resident to complete the task.-Range of motion (parameter B): The extent of movement and control exhibited by the resident while performing the task.-Panoramic view (parameter C): The ability of the resident to maintain a clear and comprehensive view of the surgical field during the task.-Conflict of instruments (parameter D): The level of efficiency demonstrated by the resident in managing instrument movements and avoiding conflicts during the task.-Exercise completed (parameter E): Whether the resident successfully completed the exercise or not in the time chosen by the simulation program.

The performance for each task was evaluated objectively according to the Hugo™ RAS simulator form, which provided a scoring system ranging from 0 to 200. Furthermore, the entire simulation performance of every trainee was assessed subjectively by the robotics tutor according to his experience, with a grade between 0 and 4, based on the attitudes of trainees, and then by a Likert scale motivation evaluation. To assess the rate of improvement, the performance of each resident was compared from their first attempt to their third attempt for each exercise, considering the aforementioned parameters, in order to achieve a global assessment. Therefore, we analyzed a tight learning curve based only on three pre-set exercises to test the new simulation platform. After completing the training, the residents compiled a 10-level Likert scale questionnaire to evaluate their overall satisfaction with the training program. The use of both objective scoring based on simulator performance and subjective evaluation by the robotics tutor allowed for a comprehensive assessment of the residents’ progress and competence in performing the robotic surgical tasks. The inclusion of the Likert scale questionnaire provided valuable feedback on the residents’ satisfaction with the training program as a whole.

### 2.3. Statistical Analysis

Continuous variables are presented as medians and interquartile ranges (IQRs) and were compared using the Wilcoxon–Mann–Whitney U test. Categorical variables are reported using frequencies and proportions and were compared using the Chi-squared test. Concerning the scores and performance in each of the three exercises, the median value of all scores was calculated and “proficiency” determined if the score of a single trainee was better than the median value of the whole trainee cohort considered for the specific exercise parameter (A, B, C, D, E). Then, a logistic regression analysis was conducted to identify variables related to the proficiency of each exercise parameter. A two-sided *p*-value of less than 0.05 was considered statistically significant. Statistical analyses were conducted using STATA (StataCorp, 2021, Stata Statistical Software: Release 17, College Station, TX, USA: StataCorp LLC).

## 3. Results

Table 1 provides pre-training data for the enrolled residents. The median age was 28.5 years (IQR, 28–30). Our cohort was composed of 16 females (36.4%) and 28 males (63.6%). Regarding specialty distribution, 12 (27.3%) participants were general surgery residents, 8 were gynaecology residents (18.2%) and 24 (54.5%) were urology residents. Concerning the year of residency, 16 participants were in their first year (36.4%), 16 in their second year (36.4%), 2 in their third year, (4.5%), 4 in their fourth year (9.1%) and 6 in their last year of residency (13.6%). In our series, 18 residents (40.9%) did not spend any time playing video games, 26 (59.1%) spent less than 3 h/day, while no one spent more than 3 h/day playing video games. Overall, 68.2% (n = 30) of the residents had prior laparoscopic exposure. Despite no residents having previous robotics experience, 90.9% (n = 40) of them expressed an interest in the field of robotic surgery.

Table 2 displays the improvements in the three exercises for all residents before and after tutorship, along with the performance for each parameter between the first and last attempts. Significant improvements were observed in most parameters for all exercises (*p* < 0.02), except for parameter D in the second exercise (*p* = 0.41), and parameters C and E in the third exercise (*p* = 0.10 and *p* = 0.41, respectively). Overall, the mean satisfaction score was 9.4 ± 1.2, indicating high satisfaction among the residents with the training program.

Regarding the proficiency score of the trainees in the three exercises:-In the first exercise, year of residency was the only potential predictor of proficiency for parameter A (OR 1.94; 95% CI [1.12–3.34], *p* = 0.01) and B (OR 1.68; 95% CI [1.02–2.27], *p* = 0.04), while for parameter D, male sex was the only predictor of proficiency (OR 0.1; 95% CI [0.02–0.43], *p* = 0.002).-In the second exercise, laparoscopic exposure was a potential predictor of proficiency for parameter C (OR 0.41; 95% CI [0.19–0.85], *p* = 0.01), while interest in robot-assisted surgery was the only predictor of proficiency for parameter E (OR 3.37; 95% CI [1.25–9.07], *p* = 0.01).-In the last exercise, interest in robot-assisted surgery was a predictor of proficiency for parameters C (OR 1.80; 95% CI [1.02–3.19], *p* = 0.04) and E (OR 3.71; 95% CI [1.59–8.62], *p* = 0.002).

## 4. Discussion

Despite the leading role of the da Vinci Surgical System in the market of robotic surgery, the rapid spread of novel robotic platforms, such as the Medtronic Hugo™ RAS System or the Cambridge Medical Robotics (CMR) Versius Surgical Robot System, has led to increasing specialization of robotic surgery in tertiary referral centres, disfavouring laparoscopic and open surgery, and discouraging low-volume institutions from approaching novel technologies [13]. In this setting, the search for new validated training curricula in robotic surgery represents a current challenge faced by both the surgical and engineering communities. Recent attempts have been made to develop a standardized robotic surgical simulation program due to growing interest in the surgical community in defining how to train surgeons, residents and medical students effectively by applying all of the emerging technologies [14].

Traditionally, since the conceptualization of surgical training programs, live animal surgery has been long utilized as a model for surgical simulation. However, bioethical concerns and cost-effectiveness implications have restricted the widespread diffusion of these training models worldwide. Moreover, animal anatomy does not accurately reflect human anatomy [15], which hampers their effectiveness as a training method. For this reason, cadaver models became the gold-standard for surgical training, due to their advantage of enabling the handling of human anatomy [16]. Nonetheless, cadaveric models have not yet been fully validated as a model for robotic surgery training. In addition, similarly to animal models, the reproducibility of cadaveric training models is limited by the scarcity of specimens and the exclusiveness of “single use”, making them less accessible for widespread training. Finally, both animal and cadaveric models do not allow for training in specific surgical procedures, such as oncological surgery, as they lack the disease pathology that is crucial for mastering specialized surgical techniques [17].

On the other hand, robotics simulation programs can provide a safe and effective environment for surgical trainees to practice and improve their skills without ethical concerns and limited resources. By utilizing technology such as virtual reality and artificial intelligence, these programs can offer tailored training experiences for various surgical procedures, including complex and specialized surgeries, thus bridging the gap left by traditional training models. By doing so, these simulation programs can better prepare surgeons and trainees, ultimately enhancing patient safety and surgical outcomes in the field of robotic surgery.

Currently, several VR simulators are available for trainees, such as the “dVSS” (Intuitive Surgical, Sunnyvale, CA, USA), the “dV-Trainer” and “Flex-VR trainer” (Mimic Technologies, Inc., Seattle, WA, USA) and the “RobotiX Mentor” (3D Systems, Littleton, CO, USA) [18], with the purpose of improving surgical skills ranging from basic to advanced, and they have the advantages of enabling trainees to familiarize themselves with the surgical console and increasing awareness and knowledge of robotic procedures. Specifically, each of these simulators may have differences in their visual resolution, components, support equipment, number of exercises available, scoring methods, system setup, security and price. In more detail, dVSS, being closely integrated with the actual da Vinci Surgical System, may offer a more seamless training experience and better alignment of skills with real surgical procedures. On the other hand, the dV-Trainer and RobotiX Mentor are independent stand-alone systems, with 3D software that replicates the actions of robotic arms in the surgical space. Unlike dVSS, they can be employed even when the robotic system is in use, allowing for flexible training opportunities for trainees. To date, proficiency in residents’ training is mainly assessed using the Global Evaluative Assessment of Robotic Skills (GEARS) metric. This tool has been shown to be able to differentiate not only experts from beginners, but also experts from intermediate surgeons [19].

In the robotic surgical landscape, the Hugo™ RAS System represents a novel promising robotic platform that is currently available in many robotic referral institutions, with applications in urological [20,21] and non-urological surgeries [22,23,24]. The spread of this new technology seems to be related to its extremely wide range of adaptability and to a fast LC in experienced robotic surgeons [25,26,27]. In detail, it is composed of an “open” surgical console with 3D passive display, which provides a three-dimensional view of the surgical field, enhancing spatial awareness; high-definition glasses with a head-tracking safety system; “*gun-like*” ergonomic hand-controllers, providing a comfortable and ergonomic grip for the surgeon; a system tower; and four independent and extendible arm-carts (Figure 4) [28]. Regarding the portable and modular system of the Hugo™ robotic platform, surgeons can move and configure it to fit and choose the best surgical approach for patients based on their features and characteristics and the type of surgical procedure, even in patients with complex physical conditions, such as increased BMI, previous abdominal surgery, anatomic kidney variability, etc. Indeed, its main technical advantages are represented by a larger working space for the bed assistant, the use of a more ergonomic trocar disposition, and cost-effectiveness. Moreover, following KARL STORZ™ technologies, the Hugo™ laparoscopic system provides three-dimensional views in full HD to better identify critical structures, a 3D HD endoscope that can be placed on any robotic arm, a Valleylab™ FT10 energy platform, a Touch Surgery™ Enterprise video management and analytics platform compatibility. Therefore, the modular configuration of the four independent arm-carts may allow for a larger and more efficient working space in the operating room, preventing unexpected system failures due to instrument clashing.

Since the recent introduction of the Hugo RAS™ system, a standardized simulator is not yet available for this platform. In this context, the aim of our program was to introduce residents to the new Hugo™ RAS system and to make them confident with it, starting from the basic skills, with the help of a specialized “robotics tutor”. An awareness of robotic devices and familiarization with troubleshooting the most common errors, such as instrument clashing, and with reproducible disruptive factors in the surgeon’s job have a great impact on their performance [28,29], should be considered the cornerstones of every robotics training program. Furthermore, Dubin et al. [29] reported a strong correlation between the presence of a human tutor during training simulation and the objectivity of robotic simulators’ evaluation of trainees’ surgical skills. Similarly, we used a two-sided evaluation of the objective results of the robotic simulator and groundbreaking subjective appraisal by an experienced tutor.

On the other hand, the most efficient tool to assess the improvement of trainees is the number of repetitions performed. Sheth et al. [30] demonstrated that a repetitive practice from the first to the tenth repetition on a robotic simulator can increase simulation skill performance (*p* < 0.001). According to the aforementioned study, we reported a significant improvement in residents’ performances (all *p* < 0.02) from the first to the last attempt for each exercise (nine repetitions), considering the five parameters evaluated. In more detail, only the “conflict of instruments” parameter in the second exercise (*p* = 0.41) and the “panoramic view” and “exercise completed” parameters in the third exercise did not highlight improvement between the first and the last attempts (*p* = 0.10 and *p* = 0.41, respectively). Therefore, the fact that the positive effect on the performance of the trainees diminished in the second and third exercises is an important observation, because this suggests that more specific and sustained training might be necessary to see further improvement in some exercises. Thus, identifying the exercises where novices struggle to improve despite initial training can help tailor training programs more effectively.

Additionally, we identified year of residency, male gender, laparoscopic exposure and interest in robotic surgery as potential predictors of proficiency in several parameters (all *p* < 0.05) in different exercises. Concerning year of residency and laparoscopic exposure, our results are in line with those reported by Mikhail et al. [31], who described a linear improvement year over year in residents continuously involved in clinical training or real laparoscopic/robotic surgery (*p* = 0.009). In clinical practice, our findings demonstrate how training programs hold the potential to diminish the learning curves of experienced open/laparoscopic surgeons or resident trainees.

Regarding gender, it is important to note that our results may be influenced by the limited and imbalanced composition of our cohort. Furthermore, the unequal distribution of males and females across different residency years adds to the potential for bias in our findings. On the other hand, interest in robotic surgery as a potential predictor may be related to the high motivation of residents to achieve basic robotic skills in order to perform robotic surgery in vivo. Sighinolfi et al. [32] reported that after robotic simulation, medical and nurse students showed great interest in robotic surgery (Hugo RAS and Versius) and some of them required for a dedicated internship (*p* < 0.001). This suggests that interest in robotics increased the positive attitudes of the trainees towards this technology.

Our study is not devoid of limitations. The sample size of our cohort may have created a bias related to the lack of identification of small differences in robotic skills at this level. Moreover, our results may be affected by the random and intentional enrolment of the residents into the program, as there was a lack of selection through inclusion and exclusion criteria. Additionally, the mentor was not blinded to the trainees’ backgrounds, which may have potentially introduced biases into the subjective evaluation. The lack of a learning curve assessment through consistent and regular training activities may have hindered the standardization of our simulation program. Furthermore, our study is the first to propose a new simulation program for the novel Hugo™ RAS platform; thus, comparisons with other validated robotics training programs may be untimely. For these reasons, further studies with a larger cohort involving more centres are needed to externally validate our initial results and enhance the overall quality of this training program, in comparison to other existing curricula. The inclusion of a broader range of participants and settings will provide a more robust foundation for drawing meaningful conclusions.

Notwithstanding these limitations, a point of strength was the inclusion of three different specialties, in order to promote the reproducibility of our model.

Finally, the overall satisfaction of residents was proven by the achievement of a 90% satisfaction rate, pointing out the high value of these simulators and the need to introduce them into standardized training programs.

## 5. Conclusions

In the current surgical landscape, the lack of validated robotic simulation programs is resulting in a heavy reliance on “in vivo” training, potentially protracting operating times and compromising the safety of surgical procedures. Thus, the standardization of a robotics training program represents an actual challenge for the future of the surgical community. Our program based on the Hugo™ RAS simulator seemed to be an effective, adaptable and reproducible method at a single institution, with significant improvements in all exercises and an overall satisfaction rate of 90% among the residents. Additionally, we found that year of residency, gender, previous laparoscopic exposure and interest in RAS were potential predictors of aptitude for robotic surgery. These findings suggest that the Hugo™ RAS system may be considered a valuable tool in the standardization of robotics training simulation programs, contributing to improving robotics skills before surgery and shortening the learning curve.

## Figures and Tables

**Figure 1 sensors-23-07348-f001:**
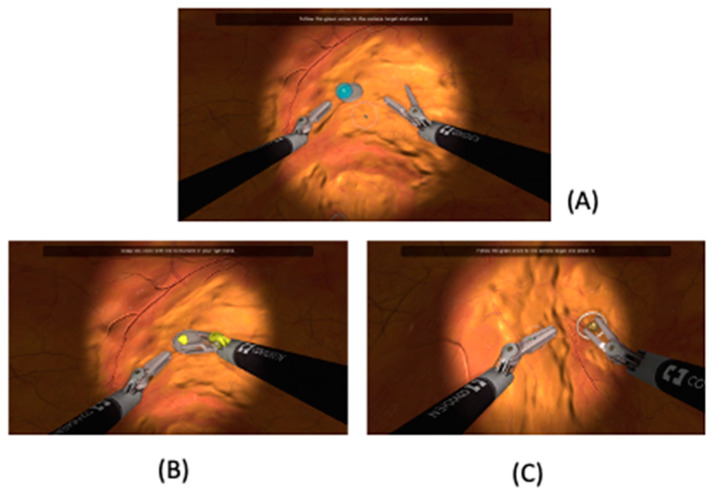
Endoscope targeting. (**A**) The trainee, using both robotic arms, moves the camera inside the virtual abdominal cavity and targets a coloured marble. (**B**) The trainee is invited to take a calculus fragment using the “Cadiere” left robotic arm. (**C**) The simulator system chooses a target distant from the current endoscope location, which must be obtained by the trainee, who must then release the calculus into a basket. In this exercise, the trainee must pay attention mainly to their visual space in order to manoeuvre the endoscope while avoiding instrument clashes. In more detail, the trainee must move the endoscope and the robotic arms alternately.

**Figure 2 sensors-23-07348-f002:**
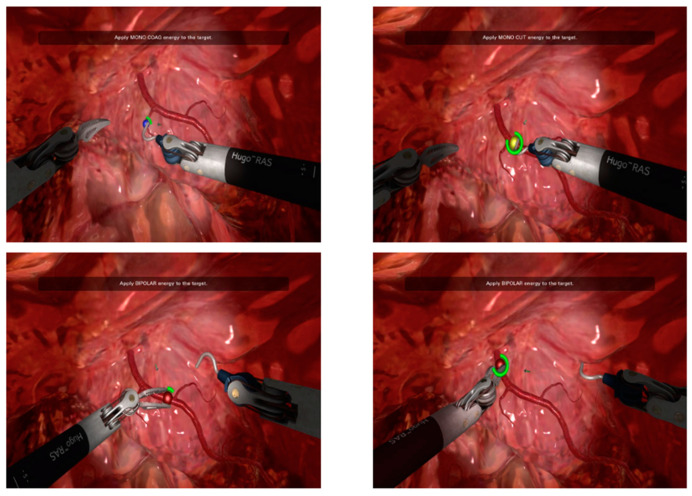
Cut and coagulate function: The trainee, moving the camera inside the abdominal space, when instructed by the simulator, will have to cut or coagulate different coloured dots placed on arterial or venous vessels of the abdomen, using the Hook or the Maryland robotic arms, respectively. In this exercise, the trainee must focus mainly on accuracy under the simulation system indications, looking for the head–foot connection to cut or coagulate in the shortest possible time.

**Figure 3 sensors-23-07348-f003:**
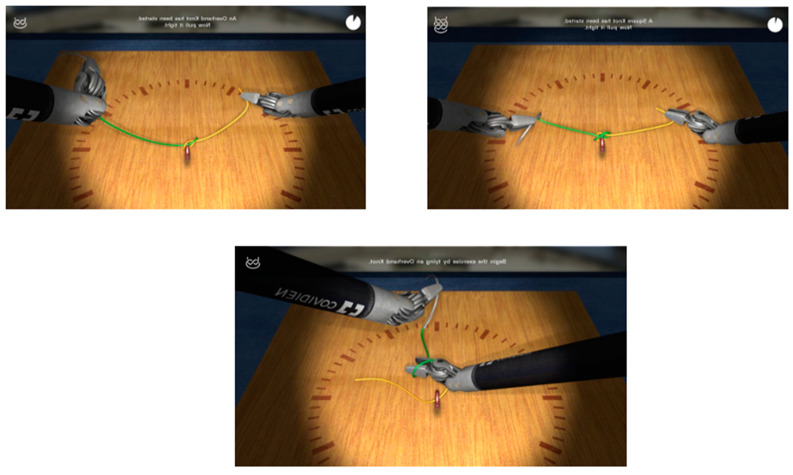
Suture skills: The trainee, using a needle driver and a Cadiere robotic arm, must perform two simple robotic knots, first in one direction, and then, in the other, while remaining inside the circle. In this exercise, the trainee must focus mainly on dexterity, trying to complete the exercise in the shortest time possible (30 s limit), while avoiding instrument clashes or exiting the boundary circle.

**Figure 4 sensors-23-07348-f004:**
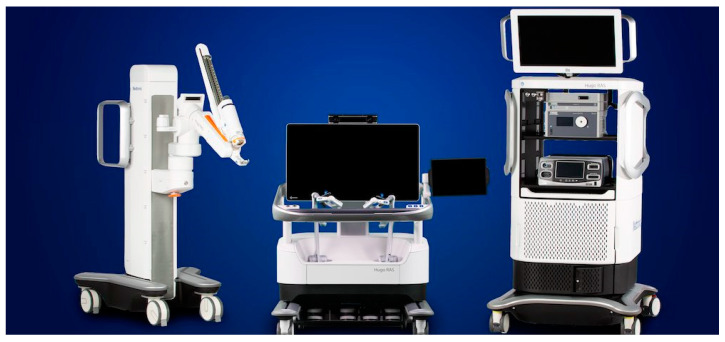
Components of the Hugo™ RAS system.

**Table 1 sensors-23-07348-t001:** Pre-training data of residents.

Pre-Training Data	Median or N (IQR or %)
**Age (years)**	28.5 (28–30)
**Gender**	
F	16 (36.4%)
M	28 (63.6%)
**Specialty**	
General surgery	12 (27.3%)
Gynaecology	8 (18.2%)
Urology	24 (54.5%)
**Year of residency**	
1	16 (36.4%)
2	16 (36.4%)
3	2 (4.5%)
4	4 (9.1%)
5	6 (13.6%)
**Time spent playing video games (hours/day)**	
0	18 (40.9%)
<3	26 (59.1%)
>3	0
**Laparoscopic exposure**	
No exposure	14 (31.8%)
Observer > 50 procedures/assistant surgeon/first surgeon	30 (68.2%)
**Robotic exposure**	
No exposure	44 (100%)
Observer > 50 procedures/bed assistant/console surgeon	
Interest in robotic surgery	0
**Interest in robotic surgery**	
0	4 (9.1%)
1	2 (4.5%)
2	12 (27.3%)
3	14 (31.8%)
4	12 (27.3%)

**Table 2 sensors-23-07348-t002:** Statistical analysis of first and last attempts for each of the three exercises and each parameter of the Hugo RAS™ system simulator.

Hugo RAS™ System Simulator Parameters (Score from 0 to 200)	Endoscope Targeting (1st Exercise)			Cut and Coagulation (2nd Exercise)			Suturing Skills (3rd Exercise)		
	First Attempt	Last Attempt	*p* *	First Attempt	Last Attempt	*p* *	First Attempt	Last Attempt	*p* *
	Median (IQR)	Median (IQR)		Median (IQR)	Median (IQR)		Median (IQR)	Median (IQR)	
**Timing (A)**	66	101.5	<0.001	93	133	0.001	0	13.5	0.001
	(23–87)	(85–115)		(85–120)	(118–144)			(0–50)	
**Range of motion (B)**	70.5	112.5	<0.001	149.5	168.5	0.001	10	88.5	0.001
	(53–99)	(91–122)		(126–162)	(162–180)		(0–79)	(21–110)	
**Panoramic view (C)**	133	200	<0.001	100	200	0.001	150	200	0.10
	(63–167)	(170–200)		(100–200)	(200–200)		(100–200)	(150–200)	
**Conflict of instruments (D)**	196	200	0.001	200	200	0.41	97.5	189.5	<0.001
	(190–200)	(199–200)		(200–200)	(200–200)		(38–182)	(85–200)	
**Exercise completed (E)**	198	200	<0.001	200	200	0.02	200	200	0.41
	(187–200)	(199–200)		(139–200)	(200–200)		(103–200)	(132–200)	

* Wilcoxon–Mann–Whitney test.

## Data Availability

This article contains no data or material other than the articles used for the review and those referenced.

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
