# Peer review of "Robotic Medtronic Hugo™ RAS System Is Now Reality: Introduction to a New Simulation Platform for Training Residents"

_sensors, 2023, doi:10.3390/s23177348_

Round 1

Reviewer 1 Report

After reading the authors' explanation of scope, I think this paper can be trusted to be published. 

Author Response

Dear Reviewer,
Thanks for the appreciations of our manuscript. 
We revised english language of our paper, as requested.

Your sincerely.

Reviewer 2 Report

In their single-center study "HugoTM RAS-system robotic-Medtronic is now reality. Introduction to a new simulation platform to train residents"  Cacciatore et al. investigate 44 residents (urology, gynaecology, general surgery) in advanced robotic simulation training with Hugo™ RAS simulator between July-September 2022. They were able to show, that almost all participators improved there parameters of 3 exercises between the first and the last attempt and that over 90% of the candidates had an interest in robotic surgery. The study is well written and clear. The topic is relevant to the readers. However, I have some minor comments.

1.  Please include Abbreviation section

2. In the discussion section the comparrisson to ‘Intuitve Surgica" should be more discussed in detail, since they are still leading the field and have most experience.

Author Response

Dear Reviewer,
Thanks for the appreciations of our manuscript and for your suggestions. 
We revised our paper, as requested, including Abbreviation section before the introduction and discussing more in detail the comparison to different simulator system (line 187-200).

“Currently, several VR simulators are now available for trainees, such as the “daVinci Skills Simulator” (dVSS Intuitive Surgical, Sunnyvale, CA), “dV-Trainer” and “Flex-VR trainer” (Mimic Technologies, Inc., Seattle, WA) and the “RobotiX Mentor'' (3D Systems, Littleton, CO) [19], with the purpose of improving surgical skills ranging from basic to advanced, including the advantages of familiarizing with the surgical console and increasing awareness and knowledge of robotic procedures. Specifically, each of these simulators may have differences in visual resolution, components, support equipment, the number of exercises available, scoring methods, system setup, security, and price. More in detail, dVSS, being closely integrated with the actual da Vinci Surgical System, may offer a more seamless training experience and better alignment of skills with real surgical procedures. On the other hand, the dV-Trainer and RobotiX Mentor are independent stand-alone systems, with a 3D software that replicates the actions of the robotic arms in the surgical space. Unlike dVSS, they can be employed even when the robotic system is in use, allowing for flexible training opportunities for trainees.”

Your sincerely

Reviewer 3 Report

In the field of Robot- Assisted Surgery, the lack of standardized robotic training courses represents an unsolved issue. In this paper, the authors aimed at testing a new training procedure in the setting of a new robotic platform, the Hugo-RAS System-Medtronic. The paper is well written and well presented. The work addresses a relevant scientific question.
I suggest to accept the manuscript with minor revisions.
In detail, the authors should:
1) fix the layout of Tables 1 and 2
2)fix the formatting of Figure 2
3)delete the Figure 4 (which is redundant, in my opinion)

Author Response

Dear Reviewer,
Thanks for the appreciations of our manuscript. 
According to your requests, we fixed figures and tables and deleted Figure 4 that was irrelevant for our paper.

Your sincerely

Reviewer 4 Report

The article is about evaluating a RAS system by means of a simulator in a study with participants. However, it is unclear whether the RAS system or the simulator is evaluated. The authors present a study with 53 subjects where abilities pre- and post-training are compared. 

There is a variety of products and brands mentioned in the article (mostly in Sections 1 and 4). However, it is unclear how these product names and brands are related to the research. Some parts of the manuscript (including the title) remind me of advertisement language.

Further, there is a large variety of abbreviations and acronyms, but a list of these is not present. Please add a list of acronyms and abbreviations for better readability.

In the study, only novices of the system under research are considered. The study applies training to these novices and tests the subjects' performance in the first and last tests. It needs to be mentioned that usually, any training in a system will have a positive effect on performance, thus a positive outcome is not surprising. As there is no comparison with another system or a benchmark, it is difficult to classify the outcomes of your study.

Regarding the evaluation, why is the GEARS metric not used for the evaluation? This could have provided the possibility to relate the results with other studies of comparable platforms or some benchmark.

The findings of the research are not compared with other results from the literature.

The study excludes subjects with a knowledge of the system under research. This means, that the study only is valid for novices. However, training activities are even more relevant for regular training activities with the system. Your research cannot address this, which is a clear limitation of your research.

In the discussion, the research is not aligned with similar research.

Several sentences and paragraphs of Section 4 are confusing:

Line 172: What is the "hierarchical" role? I don't understand the term in this context.

Line 174: What do you mean by "centralization"? And how is this related to the research?

Lines 217ff: I don't understand how this is freeing up space in the OR. Further, what do you mean by "uneventful system failures" in the context of instrument clashing?

Line 228: incomprehensible

Line 250ff: I do not follow this claim.

Line 259ff: incomprehensible

Line 270: This sounds confusing. Further, the term "paucity of robotics simulation programs" is difficult to interpret. See also Line 10.

Line 36: what is the relationship to aeronautics in the context of your research?

Figure 4 does not contribute new information. A simple list of the five parameters would suffice.

There is a variety of confusing sentences that are difficult to interpret. Occasionally, the meaning could turn into the opposite of the intended.

Lines 36/37: Something is wrong with this sentence. Risking the patient's life and minimising risks are contradictory. The sentence can be misunderstood. 

Line 48: why "Conversely"?

Line 55: "rekindle"? Is there a "d" missing?

Line 56: incomprehensible sentence.

Line 64: "the" missing

Line 184: something is wrong with this sentence; verb is missing; probably "does"

Author Response

Dear Reviewer,
Thanks for the appreciations of our manuscript and the precise suggestions in order to improve the paper’s quality. 

  1. The article is about evaluating a RAS system by means of a simulator in a study with participants. However, it is unclear whether the RAS system or the simulator is evaluated. The authors present a study with 53 subjects where abilities pre- and post-training are compared. There is a variety of products and brands mentioned in the article (mostly in Sections 1 and 4). However, it is unclear how these product names and brands are related to the research. Some parts of the manuscript (including the title) remind me of advertisement language.

  • Thank you for your feedback on the manuscript. We appreciate your observations and acknowledge that the inclusion of multiple product names and brands may give the impression of advertisement language in certain parts of the manuscript, including the title, or in Section 1 and 4. We apologize, but our aim was to provide more information to those who are new to the field of simulators and increase readers' interest. Additionally, we sought to bolster the validity of our work by making comparisons with previously simulators or robotic curricula published in the literature.
  1. Further, there is a large variety of abbreviations and acronyms, but a list of these is not present. Please add a list of acronyms and abbreviations for better readability.
  • Thank you for the advice. We added Abbreviation section as requested.
  1. In the study, only novices of the system under research are considered. The study applies training to these novices and tests the subjects' performance in the first and last tests. It needs to be mentioned that usually, any training in a system will have a positive effect on performance, thus a positive outcome is not surprising. As there is no comparison with another system or a benchmark, it is difficult to classify the outcomes of your study.
  • Thanks to your feedback in the field of our results. Although the improvement in all fields may not seem surprising at first, our analysis not shows positive effect in all fields. Indeed, the positive effect on performance is greater in the first exercise. The fact that the positive effect on performance diminishes in the second and third exercises (cut and coagulate and nodes) is an important observation, because 1 and 2 parameters respectively, are not significant (p > 0.05). Therefore, this fact suggests that more specific and sustained training might be necessary to see further improvements in some exercises. Identifying the exercises where novices struggle to improve despite initial training can help tailor training programs more effectively. We added this observation also in the discussion (line 281-286).
  1. Regarding the evaluation, why is the GEARS metric not used for the evaluation? This could have provided the possibility to relate the results with other studies of comparable platforms or some benchmark.
  • Thanks for your clarification. Regarding the evaluation, we couldn’t consider the entire GEARS metric, because we used the pre-set Hugo RAS simulation platform, which has some parameters in common and some different, comparing the parameters with the GEARS.
  1. The findings of the research are not compared with other results from the literature.
  • We added a comparison between our results and DaVinci and Versius simulators (line 287-294)
  1. The study excludes subjects with a knowledge of the system under research. This means, that the study only is valid for novices. However, training activities are even more relevant for regular training activities with the system. Your research cannot address this, which is a clear limitation of your research.
  • Thanks for the advisement. Our intent was not to evaluate the learning curves of experienced surgeons, but rather to test a new simulation platform using the new Hugo RAS robotic system. Moreover, we added this sentence as a clear limitation of our research. (line 306-314)
  1. In the discussion, the research is not aligned with similar research.
  • We cited and discussed other similar research in the field of robotic simulator (Line 287-294 and line 300-305)
  1. Several sentences and paragraphs of Section 4 are confusing:

Line 172: What is the "hierarchical" role? I don't understand the term in this context.

Line 174: What do you mean by "centralization"? And how is this related to the research?

Lines 217ff: I don't understand how this is freeing up space in the OR. Further, what do you mean by "uneventful system failures" in the context of instrument clashing?

Line 228: incomprehensible

Line 250ff: I do not follow this claim.

Line 259ff: incomprehensible

Line 270: This sounds confusing. Further, the term "paucity of robotics simulation programs" is difficult to interpret. See also Line 10.

Line 36: what is the relationship to aeronautics in the context of your research?

Figure 4 does not contribute new information. A simple list of the five parameters would suffice.

  • Thanks to your attention and corrections. We revised and modified all sentences highlighted, as requested. Additionally, we deleted Figure 4 and the relationship with aeronautics.
  1. Comments on the Quality of English Language

There is a variety of confusing sentences that are difficult to interpret. Occasionally, the meaning could turn into the opposite of the intended.

Lines 36/37: Something is wrong with this sentence. Risking the patient's life and minimising risks are contradictory. The sentence can be misunderstood.

Line 48: why "Conversely"?

Line 55: "rekindle"? Is there a "d" missing?

Line 56: incomprehensible sentence.

Line 64: "the" missing

Line 184: something is wrong with this sentence; verb is missing; probably "does"

  • Thanks for your advisement. We edited the quality of English grammar and modified the sentences as requested.

Your sincerely,
Dott.ssa Manuela Costantini
IRCSS “Regina Elena” National Cancer Institute
Via Elio Chianesi, 53, Roma, 00128

Round 2

Reviewer 4 Report

The authors have improved their manuscript, and some parts have become clearer. The detailed description of the metrics is valuable. However, there are still several issues that need to be addressed before publication.

The contributions to the state of the art in the field seem rather weak. The authors claim that their manuscript describes the first study for a certain product, but no comparisons of their results with other results (for the other products) are given. 

The discussion mentions other products with similar functionality. However, the results from the study do not have any impact on the text in the discussion section. Thus, parts of the description of these other products should be moved to the background section.

The discussion section should contain a discussion of the results, and what they mean; including which impact they might have.

In the results, several correlations are mentioned, e.g., that gender or interest in RAS would be predictors. What does this mean in practice? Would that imply that the training system is only suited for one gender or for those who are interested in RAS? What would be the mechanisms behind this? This needs to be further discussed.

Further, it seems that a search for correlations was performed. The multiple comparisons problem might be an issue here. Please check this. Please, also note that finding a correlation is not very valuable as long as there is no explanation for this.

The conclusion section is still very generic. Only a very overall statement about the effectiveness of the RAS system is given.

There are some terms that seem confusing. The term "trainer" seems to be wrongly used. A trainer is a person who trains someone. Probably, the term "trainee" is more correct in your manuscript.

Further, the term "robotic tutor" (e.g. Line 21) is used. I would interpret a robotic tutor as a robot that teaches a person. However, it seems that you mean something else.

Line 52: I don't understand the contents of this sentence. What role does the COVID-19 pandemic play in this context? What do you mean by "rarity ..."? Do you mean "Because of the pandemic, robotic simulation programmes have become rare"? Still, I don't understand this.

The results of Table 2 are just presented, but not discussed.

paragraph starting Line 214: generic and not related to the findings of the study.

Line 282ff: incomprehensible.

I still don't understand your concept of how to evaluate the RAS system. Please explain the methodology of the evaluation prior to the presentation of the results. I see from Table 2 that there is some progress for the participants of the study, but it seems unclear how much comes from the concrete implementation and how much would be generic (e.g., that the simulator of competing systems would result in a similar learning curve).

paragraph Line 301: Incomprehensible. Do you mean: Because they participated in the training, they became interested in the technology, with the consequence that they are successful (ref. predictor)."? I don't quite get this.

Line 312: incomprehensible sentence.

The text is still hard to read, and many sentences are confusing. On several occasions, the opposite of what the authors probably mean could be read out of the presented text. There are several passages and sentences that are incomprehensible.

Line 19: interested in robotic -> interested in robotics

Line 52: I really don't understand this sentence.

In Table 1, why are the years of residency in Roman numbers? This looks odd. Please revise.

Line 282: something is wrong with this sentence.

Line 320: incomprehensible.

Author Response

The authors have improved their manuscript, and some parts have become clearer. The detailed description of the metrics is valuable. However, there are still several issues that need to be addressed before publication.

  1. The contributions to the state of the art in the field seem rather weak. The authors claim that their manuscript describes the first study for a certain product, but no comparisons of their results with other results (for the other products) are given. 

  • Thanks to for the accurate comments. We added the lack of real comparisons between training in our limitations section and we explained why in our opinion is to early to make comparisons with the other existing curricula (Line 315-321).

  1. The discussion mentions other products with similar functionality. However, the results from the study do not have any impact on the text in the discussion section. Thus, parts of the description of these other products should be moved to the background section.

  • Following your requests, we moved the discussion section concerning other existing training programs to the background section (Line 60-64)

  1. The discussion section should contain a discussion of the results, and what they mean; including which impact they might have.

  • Thanks for the precise comments. We added some lines on the potential role of our findings in clinical practice, as well as their impact on experienced open/laparoscopic surgeon or resident (Line 286-299)

  1. In the results, several correlations are mentioned, e.g., that gender or interest in RAS would be predictors. What does this mean in practice? Would that imply that the training system is only suited for one gender or for those who are interested in RAS? What would be the mechanisms behind this? This needs to be further discussed.

-          Thanks for the advisement. Our intent was to analyse the proficiency score of the trainees in the three exercises. We found year of residency, laparoscopic exposure, gender and interest in robotics as potential predictors of proficiency. We tried to explain these results with a comparison with other previous studies. Concerning gender, our results may be affected by the restricted uneven cohort. In fact male and female are not equally distributed in the year of residency, enhancing the potential bias of these results (Line 292-309)

  1. Further, it seems that a search for correlations was performed. The multiple comparisons problem might be an issue here. Please check this. Please, also note that finding a correlation is not very valuable as long as there is no explanation for this.

  • Thank you for your feedback on the manuscript. We appreciate your observations and acknowledge that a novel training program needed multiple comparisons. Nonetheless, our purpose was to provide more information to the new robotic system and in our opinion the comparisons with other validated simulators is premature, as explained in our limitations (Line 313-319)

  1. The conclusion section is still very generic. Only a very overall statement about the effectiveness of the RAS system is given.

  • Thanks to your feedback in the field of our conclusions. According your request, we modified this section introducing more detailed information on overall satisfaction and effectiveness (Line 328-339)

  1. There are some terms that seem confusing. The term "trainer" seems to be wrongly used. A trainer is a person who trains someone. Probably, the term "trainee" is more correct in your manuscript. Further, the term "robotic tutor" (e.g. Line 21) is used. I would interpret a robotic tutor as a robot that teaches a person. However, it seems that you mean something else.

  • Thanks to your recommendations. We revised the two terms “trainer” and “robotic tutor” as requested.

  1. Line 52: I don't understand the contents of this sentence. What role does the COVID-19 pandemic play in this context? What do you mean by "rarity ..."? Do you mean "Because of the pandemic, robotic simulation programmes have become rare"? Still, I don't understand this.

-          Thanks to your precisation, we noticed that this sentence had no proper context in our paper. Therefore, we deleted it with corrispective reference.

  1. The results of Table 2 are just presented, but not discussed.

  • We discussed the results of Table 2 in the discussion section (Line 277-287), as requested.

  1. Paragraph starting Line 214: generic and not related to the findings of the study.
  2. Line 282ff: incomprehensible.

  • Thanks to your advisement. We cancelled and revised the line 214 and 282, respectively.

  1. I still don't understand your concept of how to evaluate the RAS system. Please explain the methodology of the evaluation prior to the presentation of the results. I see from Table 2 that there is some progress for the participants of the study, but it seems unclear how much comes from the concrete implementation and how much would be generic (e.g., that the simulator of competing systems would result in a similar learning curve).

  • Thanks to your requests. We explained more in detail the methodology of evaluation in the material and methods section (Line 114-118), considering the progress for the residents.

  1. paragraph Line 301: Incomprehensible. Do you mean: Because they participated in the training, they became interested in the technology, with the consequence that they are successful (ref. predictor)."? I don't quite get this.
  2. Line 312: incomprehensible sentence.

  • We revised the sentences (line 302-307 and line 313-315, respectively) in order to make it more comprehensible for the readers.

Comments on the Quality of English Language

The text is still hard to read, and many sentences are confusing. On several occasions, the opposite of what the authors probably mean could be read out of the presented text. There are several passages and sentences that are incomprehensible.

Line 19: interested in robotic -> interested in robotics

Line 52: I really don't understand this sentence.

In Table 1, why are the years of residency in Roman numbers? This looks odd. Please revise.

Line 282: something is wrong with this sentence.

Line 320: incomprehensible.

  • Thanks for your accurate advisements. We edited the quality of English grammar and modified or deleted the sentences as requested.

Your sincerely,

Manuela Costantini